# EnDPoINT: protocol for an implementation research study to integrate a holistic package of physical health, mental health and psychosocial care for podoconiosis, lymphatic filariasis and leprosy into routine health services in Ethiopia

Maya Semrau ![ORCID],[1] Oumer Ali,[1,2] Kebede Deribe,[1,3] Asrat Mengiste,[2] Abraham Tesfaye,[2] Mersha Kinfe,[2] Stephen A Bremner,[4] Natalia Hounsome,[1] Louise A Kelly-Hope,[5] Hayley MacGregor,[6] Henock B Taddese,[7] Hailom Banteyerga,[8] Damen HaileMariam,[3] Nebiyu Negussu,[9] Abebaw Fekadu ![ORCID],[1,2] Gail Davey[1]

For numbered affiliations see end of article.

**Correspondence to**
Dr Maya Semrau;
m.semrau@bsms.ac.uk

## ABSTRACT

**Introduction** Neglected tropical diseases (NTDs) causing lower limb lymphoedema such as podoconiosis, lymphatic filariasis (LF) and leprosy are common in Ethiopia. Routine health services for morbidity management and disability prevention (MMDP) of lymphoedema caused by these conditions are still lacking, even though it imposes a huge burden on affected individuals and their communities in terms of physical and mental health, and psychosocial and economic outcomes. This calls for an integrated, holistic approach to MMDP across these three diseases.

**Methods and analysis** The 'Excellence in Disability Prevention Integrated across NTDs' (EnDPoINT) implementation research study aims to assess the integration and scale-up of a holistic package of care—including physical health, mental health and psychosocial care—into routine health services for people with lymphoedema caused by podoconiosis, LF and leprosy in selected districts in Awi zone in the North–West of Ethiopia. The study is being carried out over three phases using a wide range of mixed methodologies. Phase 1 involves the development of a comprehensive holistic care package and strategies for its integration into the routine health services across the three diseases, and to examine the factors that influence integration and the roles of key health system actors. Phase 2 involves a pilot study conducted in one subdistrict in Awi zone, to establish the care package's adoption, feasibility, acceptability, fidelity, potential effectiveness, its readiness for scale-up, costs of the interventions and the suitability of the training and training materials. Phase 3 involves scale-up of the care package in three whole districts, as well as its evaluation in regard to coverage, implementation, clinical (physical health, mental health and psychosocial) and economic outcomes.

### Strengths and limitations of this study

► The Excellence in Disability Prevention Integrated across NTDs (EnDPoINT) implementation research study takes a holistic approach in that it explores the full scope of processes and outcomes involved in the integration of physical health, mental health and psychosocial care for people with lower limb lymphoedema caused by podoconiosis, lymphatic filariasis and leprosy; integration of care relates to: (1) integration of care across the three diseases; (2) integration of care for these diseases into routine health services and (3) integration of mental health and psychosocial (including stigma reduction) interventions into holistic packages of care.

► EnDPoINT is guided by relevant conceptual perspectives on implementation research as well as employing a 'Theory of Change' approach towards devising, managing and evaluating the integration study.

► The study uses a wide range of mixed-method approaches across three phases to develop, scale-up and evaluate the holistic care package.

► Limitations of the study are: selection of study districts partly based on accessibility; time and resource constraints that do not allow the pilot study to have a control group or full-length follow-up before the scale-up; possible contamination of the study results by previous or ongoing work of other organisations in the study zone.

**Ethics and dissemination** Ethics approval for the study has been obtained in the UK and Ethiopia. The results will be disseminated through publications in scientific journals, conference presentations, policy briefs and workshops.

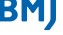

**Strengths and limitations of this study**

► The study team comprises an array of multidisciplinary academics and professionals in Ethiopia as well as internationally, including medics, epidemiologists, psychiatrists, policy experts, social scientists, participatory intervention methodologists and affected persons.

## INTRODUCTION

Neglected tropical diseases (NTDs) that result in lower limb lymphoedema, that is, swelling of the lower leg, include podoconiosis, lymphatic filariasis (LF) and leprosy. These conditions are common in Ethiopia, with an estimated 1.53 million cases of podoconiosis in the country,[1] and around 300 000 people affected by leprosy.[2] The exact prevalence for LF lymphoedema in Ethiopia is less well known, since recent mapping studies have resulted in differing estimates[3 4]; however, integrated morbidity mapping in 2018 in 20 podoconiosis–LF coendemic districts identified a prevalence of 84.9 per 10 000 population (26 123 cases in total), of whom 95.3% had leg lymphoedema only, 2.9% had hydrocele (swelling of the genitals in men), 1.5% had both leg lymphoedema and hydrocele and 0.3% cases had breast lymphoedema.[5] Nationwide mapping in 2013 demonstrated that podoconiosis accounts for approximately 64.8%, LF for 13.2% and leprosy for 12.8% of the total burden of lymphoedema in Ethiopia.[6] With global burden estimates of 15 million, 4 million and 2–3 million for LF lymphoedema, podoconiosis and people affected by leprosy, respectively,[7–9] improving morbidity management and disability prevention (MMDP) and enhancing our understanding of processes and outcomes of these efforts, in particular how to integrate MMDP into existing routine primary community-based healthcare services, are likely to be of global significance and address resource and sustainability questions.[10]

Lymphoedema imposes huge burdens on affected individuals and their communities in terms of disability such as reduced mobility and pain,[11] mental distress,[12] depression and anxiety,[13 14] stigma, discrimination and social exclusion,[11 15–18] which can limit health-seeking behaviours and access to social services including education, thereby leading to loss of economic productivity at household, community and national level.[19–22] For podoconiosis in Ethiopia alone, affected people lose on average 45% of their economically productive time due to morbidity associated with the disease,[20] and the estimated cost to Ethiopia's economy is US$213 million per year.[23] These conditions therefore compromise the livelihoods and well-being of populations that are already disadvantaged and hard-to-reach, for example, because they live remotely.[24] Burden of disease measures are even higher when taking into account the psychological and emotional consequences of NTDs, with estimates that the burden of these psychosocial outcomes may be double that of the physical health consequences.[25]

Along with the mental distress and disorder that commonly accompanies these NTDs, stigma is another key issue that significantly increases the disease burden for these diseases,[25] and which acts as major barrier to accessing MMDP services.[26 27] It is now widely recognised that joint approaches to reduce stigmatisation across NTDs may be feasible given the similarities in causes, manifestations and interventions,[15] but there remains a knowledge gap in regards to relevant, evidence-based stigma reduction interventions for use within integrated MMDP programmes. Prior research suggests that stigma for lower limb lymphoedema has three main causes: (1) misinformation among the community, affected persons and their families about the diseases' causes, treatment and prevention,[16 28] which could be addressed through educational interventions providing standardised health information, to increase disease-related health literacy; (2) the common poverty and reduced quality of life due to affected individuals' lost economic productivity,[29] which could be addressed through community-based socioeconomic rehabilitation/strengthening of affected individuals and their families and (3) the economic burden related to the costs of care, including transport to health facilities,[16 30] which could be lessened by providing integrated services in nearby health facilities at low or no cost. Since multicomponent interventions are more effective than single-component interventions for stigma reduction, it is important for programmes to address all three causes of stigma (which the project described here attempts to do).

The WHO has published targets for elimination of many NTDs, including LF and leprosy[31 32]; however, this has not included podoconiosis (though podoconiosis has been included under LF for MMDP). While foot care for leprosy patients has been integrated into routine health services in Ethiopia since 2001,[33] foot care interventions for podoconiosis and LF aiming to prevent disability are currently mainly provided through donor-supported projects in a disease-specific and disparate manner. Knowledge of these neglected diseases within healthcare systems is therefore often inadequate, diagnosis and treatment options are limited, and the cost of accessing healthcare to ease symptoms can be prohibitively expensive. On the other hand, the Ethiopian Federal Ministry of Health (FMOH) has developed integrated guidelines for LF and podoconiosis MMDP but these did not include leprosy (nor did they include mental health or psychosocial components). Yet, there is a clear rationale for the integration of care across these three conditions because of the shared clinical symptomatology of lower leg lymphoedema. In addition, as the mental distress and illness that commonly accompanies lymphoedema[12 13 34] often go untreated,[35] care can be enhanced further for affected people by adding mental health and psychosocial components to physical health services. Integration of care for lymphoedema can therefore include three aspects:

(1) integration of care across the three diseases; (2) integration of care for these diseases into routine health services and (3) integration of mental health and psychosocial (including stigma reduction) interventions into holistic packages of care. Integrated care models for NTDs have also been used in other countries.[36 37]

The Ethiopian FMOH has recognised the importance of these issues, and as a result, both podoconiosis and LF have now been included in the first two National Master Plans (2013–2015 and 2015/2016–2019/2020) for integrated control of NTDs[38 39] and identified as two of eight priority NTDs in the country, while at the same time aiming to move away from vertical programmes. There have also been calls to improve the responsiveness of the health system and include mental health and psychosocial care into the MMDP of podoconiosis and other lymphoedema in Ethiopia.[12] The Health Extension Programme, rolled out successfully in Ethiopia since 2003, now boasts more than 38 000 community-based health extension workers (HEWs), and a supervisory system to support them.[40] Their reach has been extended through the women-centred Health Development Army, members of which link 'model families' with five other households to implement health initiatives.[30] These cadres are ideally placed to offer simple, low-tech foot care as well as mental health and psychosocial support to people living with lymphoedema in Ethiopia,[41] and to refer on people who need more specialised care.

To address these issues and make use of the opportunities that currently exist in Ethiopia, the FMOH requested implementation research to guide integration of a holistic care package, including physical health, mental health and psychosocial care interventions, for people with lower-limb lymphoedema into government-run health services. The 'Excellence in Disability Prevention Integrated across NTDs' (EnDPoINT) research study (2017–2021), funded by the National Institute for Health Research in the UK, was set up in response to this and our findings are expected to inform existing plans for scale-up of integrated physical limb care and mental health and psychosocial support interventions across Ethiopia. The protocol for the EnDPoINT study is outlined below.

## METHODS AND ANALYSIS
### Conceptual framework for study
EnDPoINT is an implementation research study in that its research questions have a strong focus on implementation strategies, it includes a wide range of implementation measures, and it is being conducted in real-world settings and populations.[42] The study takes guidance both from the Context and Implementation of Complex Interventions (CICI) framework, as well as from the Medical Research Council (MRC) framework for development and evaluation of complex interventions,[43 44] with further explication and reinforcement of the different steps

within the MRC framework through a 'Theory of Change' (ToC) approach.[45]

The CICI is a comprehensive framework that incorporates implementation, context and setting as dimensions, and interventions are seen to be operationalised within the macrolevel, mesolevel and microlevel. A core interest of EnDPoINT is to generate transferable learning as to the mechanisms at work during integration of targeted interventions into health systems within low-income and middle-income countries. As such, the study is a theory-led investigation, for which the CICI can provide a useful framework. We hope to generate rich descriptions of processes and mechanisms of effect, within the Ethiopian context, thereby enabling inference and transferability to similar settings and conditions.

The MRC framework for complex interventions provides a model for the practical steps involved in the development and evaluation of EnDPoINT's care package (sitting within the broader theoretical framework provided by the CICI). The MRC framework has been used widely and proposes four phases in the development and evaluation of interventions: (1) development, (2) feasibility/piloting, (3) evaluation, and (4) implementation; these phases are considered to be an iterative rather than a linear process, and can facilitate locally appropriate strategies to be generated. ToC is a structured thinking process to build a 'causal pathway' of what needs to change and why, in order to achieve the desired outcomes[45]; it enhances the MRC framework for complex interventions through its emphasis on a theory-driven approach, which can be incorporated into and provide practical guidance for the different phases of the MRC framework.[45] Figure 1 shows how ToC may theoretically link in and strengthen each of the four phases of the MRC framework on complex interventions. Within the EnDPoINT study, the ToC approach will be used within two of the four phases of the MRC framework: the development and feasibility/piloting phases; for these two phases, there is good evidence for the practical viability and usefulness of an embedded ToC approach.[46]

### Study design
EnDPoINT's aims are to facilitate effective access to physical health, mental health and psychosocial care for people with lower limb lymphoedema caused by podoconiosis, LF and leprosy in selected districts in Awi zone in Ethiopia through integration and scale-up of a holistic care package into government-run health services.

The study is organised according to three phases, which map onto the different phases of the MRC framework for complex interventions.[43 44] It takes an iterative approach, whereby the subsequent phases are dependent on findings made during the previous phases. See table 1 for the objectives, research questions, research activities/methods and outcome measures for each of the three phases.

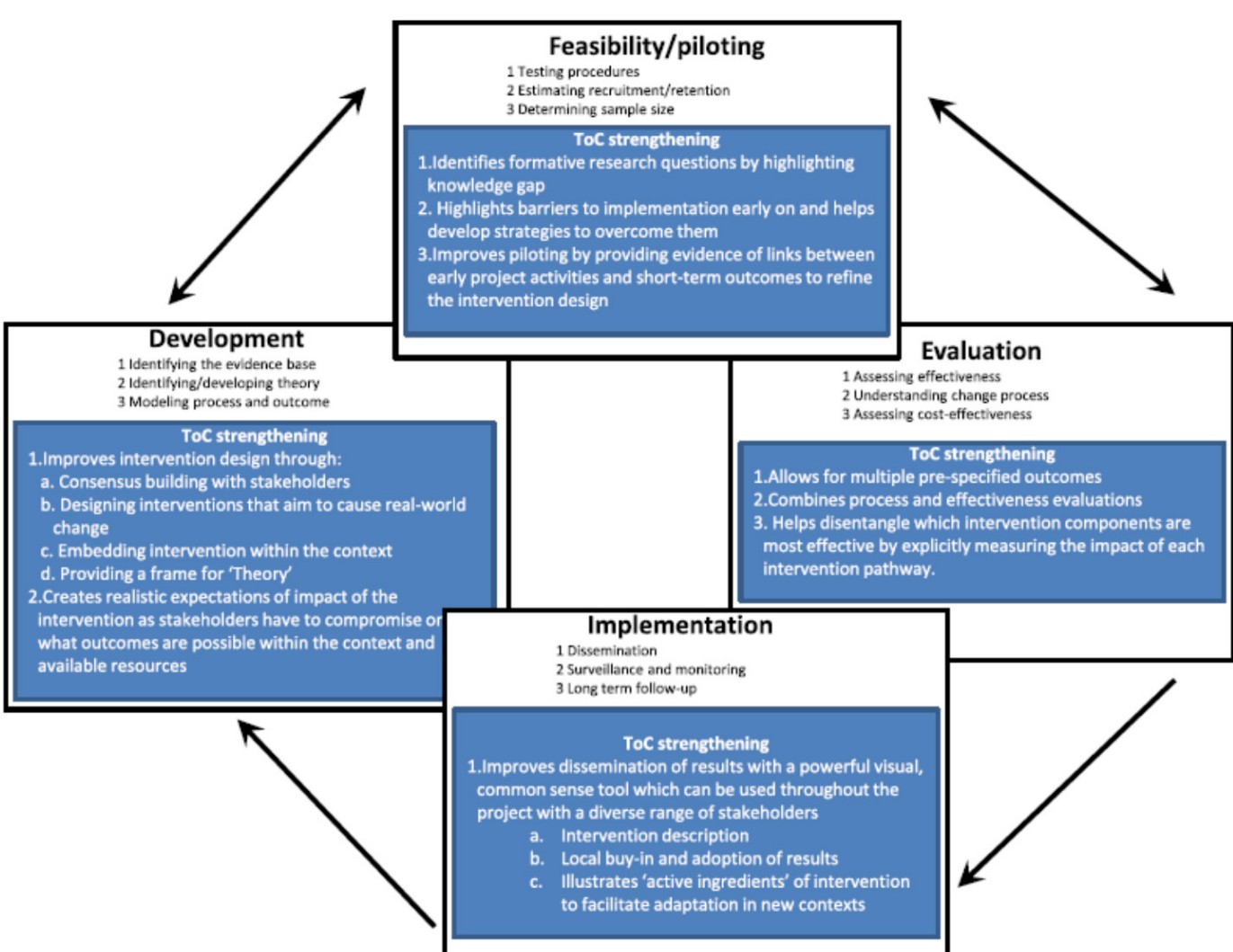

**Figure 1** 'Theory of Change' (ToC) within the MRC framework for complex interventions (taken unchanged from de Silva *et al*[45]) (in white: taken directly from MRC framework; in blue: added elements on ToC).

Phase 1 of EnDPoINT (months 1–20 of the project)—which corresponds to phase 1 of the MRC framework—entails various research activities to inform the development of the holistic care package. More specifically, the EnDPoINT care package builds on an MMDP (physical) self-care package that has previously been established and tested in Ethiopia for podoconiosis,[47] but with the addition of mental health and psychosocial components, as well as enabling provision of integrated care across the three diseases. The development of the care package within phase 1 thereby relates to: (1) the identification and design of these added mental health and psychosocial elements, while (2) integrating care across the three conditions, as well as (3) identifying strategies for integration of the care package into the government-run healthcare delivery system (as MMDP services are not yet routinely provided in Ethiopia for podoconiosis and LF). A bottom-up approach is used for this, taking into account a wide range of stakeholder views within multiple fora (see table 1).

Phase 2 (months 21–30)—which corresponds to phase 2 of the MRC framework—involves a pilot study of the holistic care package in one subdistrict in Awi zone, which includes research activities to assess the care package's adoption, feasibility, potential effectiveness and fidelity when integrated into government-run health services and across the three diseases, as well as the observable trends in the utilisation and coverage of the care package.

Phase 3 (months 31–42)—which corresponds to phases 3 and 4 of the MRC framework—entails the scale-up of the holistic care package of the integrated three disease MMDP embedded into the primary healthcare facilities, and its evaluation in three selected districts in Awi zone.

### Sample selection

Awi zone, in which the EnDPoINT study is being conducted, is 1 of 10 zones in the Amhara Region in the North-West of Ethiopia and is divided into 12 districts (or *woredas*). Figure 2 shows the three districts selected for the study: one subdistrict (a cluster that includes five villages) in Guagusa Shikudad district for piloting

**Table 1** Objectives, research questions, research activities/methods and outcome measures for each of the three phases within the Excellence in Disability Prevention Integrated across NTDs (EnDPoINT) study

| | Objectives | Research questions | Research activities/methods | Outcome measures |
|---|---|---|---|---|
| Phase 1: Development of care package | ▲ Finalise a comprehensive package of holistic physical health, mental health and psychosocial care for people with podoconiosis, LF and leprosy<br><br>▲ Learn lessons about integration from implementers and beneficiaries of care to date<br><br>▲ Develop strategies for integrating and evaluating the holistic care package into the routine healthcare delivery system in selected districts in Awi zone in Ethiopia, and integrating care across the three diseases | ▲ What are the key elements that constitute optimal physical health, mental health and psychosocial care for people with podoconiosis, LF and leprosy?<br><br>▲ What strategies need to be developed to facilitate integration of the holistic care package into the routine healthcare delivery system and across the three diseases?<br><br>▲ What are the critical contextual factors (including drivers and barriers) that influence the process of integration of the holistic care package into government-run health services and across the three diseases?<br><br>▲ What are the key features of the intervention that influence the manner of integration into the healthcare system and across the three diseases?<br><br>▲ Who are the key health system actors that have a stake in the integration of the care package into the government-run health services, and what coordination and capacity building needs exist?<br><br>▲ Is the draft care package feasible, acceptable and appropriate in terms of its integration into government-run health services? | ▲ Document review of grey literature, including existing national NTD guidelines, other relevant documents on care provision for NTDs and/or mental health, study reports and programme documents, to inform and guide the development of the care package.<br><br>▲ Systematic review of publications in scientific journals, on the functional/disability, mental health and psychosocial outcomes associated with podoconiosis, LF and leprosy, to complement the document review.<br><br>▲ Situational analysis/resource mapping, to collect cross-sectional baseline data on contextual factors relevant to the development, implementation and integration of the care package, as well as to identify any resources available for this, and potential risk factors.<br><br>▲ Three 'Theory of Change' (ToC) workshops with members of the research team and key stakeholders, to identify and establish the key causal pathways between the desired outcomes, interventions, assumptions, indicators and measurement of the outcomes for the care package (represented visually through a ToC map), as well as to encourage stakeholder buy-in to the study.[45]<br><br>▲ Key informant interviews and focus group discussions with stakeholders, to assess the draft care package's feasibility (ie, the extent to which the intervention can be carried out within the routine health system), acceptability (ie, the perception among stakeholders that the care package is agreeable), and appropriateness (ie, the perceived fit or relevance of the care package to key stakeholders), and to assess key aspects of the ToC (for example, assumptions made).<br><br>▲ Workshop with key stakeholders to discuss the draft care plan and training materials that will be adapted.<br><br>▲ Informed by the above steps, *develop* a comprehensive care package consisting of interventions at the healthcare organisation/coordination, health facility and community level. | Qualitatively assessed implementation outcomes, including feasibility, acceptability and appropriateness. |

Continued

**Table 1** Continued

| | Objectives | Research questions | Research activities/methods | Outcome measures |
|---|---|---|---|---|
| Phase 2: Piloting of care package in one subdistrict | ▲ Implement and evaluate the care package in one subdistrict in Awi zone in Ethiopia<br>▲ Develop a monitoring and evaluation plan for the subsequent scale-up of implementation of the care package | The research questions outlined for phase 1 are sustained in Phase 2. Additional research questions during phase 2 are:<br>▲ Is the holistic care package adoptable, feasible, potentially effective, and of high fidelity when integrated into government-run health services and across the three diseases?<br>▲ What are the observable trends in the utilisation and coverage of the care package? | ▲ Pilot study of the care package in one subdistrict in Awi zone, to assess its adoption (ie, the intention of trying to employ the care package), feasibility, acceptability, fidelity (ie, the degree to which the care package was implemented as designed), effectiveness (ie, the impact of the care package as delivered on individual patient outcomes), costing of the care package, its readiness for scale-up, and the suitability of the training and training materials. This is achieved through:<br> – Observation<br> – Key informant interviews and/or focus group discussions with key stakeholders such as people who received the training and/or those who delivered or received the interventions; cost data will be collected from programme managers.<br> – Before-and-after (pre/post) collection of quantitative data, including number of cases identified and treated (and whether there were any differences in the way these were identified), patient-level outcomes, and for the training 'change of knowledge, attitudes and practice' (KAP) and satisfaction questionnaires<br>▲ Workshop with the NTD Department of the Ethiopian FMOH<br>▲ Based on the pilot study and workshop, development of a protocol to evaluate scale-up of integration of the holistic care package across the three diseases into the government-run health system in three districts in Awi zone in Ethiopia. | ▲ Qualitatively assessed implementation outcomes, including adoption, feasibility, acceptability and fidelity.<br>▲ Effectiveness of care package interventions assessed at baseline and 3-month follow-up, indicated by number of cases identified and treated, and patient-level outcomes (pre/post) through structured questionnaires, including MMDP assessment, swelling circumference, frequency of acute attacks, stage and grade of affected limb, depression (measured through the Patient Health Questionnaire 9, PHQ-9), suicidal ideation (CIDI), alcohol use (FAST), quality of life (Dermatology Quality of Life Index, DQLI), disability (WHODAS 2.0), internalised stigma (ISRL), discrimination (DISC-12), social distance (Social Distance Scale, SDS), social support (OSLO 3), happiness index, and explanatory models.<br>▲ Training evaluations, measured by questionnaires on 'change of knowledge, attitudes and practice' (KAP) (immediately before and after training), and mixed-method satisfaction data (immediately after training).<br>▲ Cost of interventions (economic outcomes). |
| Phase 3: Scale-up and evaluation of care package in three districts | ▲ Scale up the holistic care package across the three diseases into government-run health services in three districts in Awi zone in Ethiopia based on the findings from the pilot study in the single subdistrict (during phase 2)<br>▲ Evaluate the scale-up of the care package<br>▲ Conduct analysis of the intervention costs during the scale-up | ▲ What are the critical factors (including drivers and barriers) that influence the process of scaling-up the care package, and that ensure its effectiveness, sustainability, quality and coverage?<br>▲ How does the context interact with the intervention to influence the effectiveness of integration, that is, how do these elements fare in the different contexts presented by the three districts?<br>▲ Does the care package result in improved outcomes for people with podoconiosis, LF and leprosy, including clinical (physical health, mental health and psychosocial), economic and social outcomes?<br>▲ What is the economic impact of the care package? | ▲ Implementation of the interventions that are included in the care package and that were developed and piloted during phases 1 and 2 of the study. These will include interventions at the healthcare organisation/coordination, health facility and community levels.<br>▲ Evaluation of care package through mixed methodologies, that is:<br> – Before-and-after (pre/post) collection of quantitative data, including number of cases identified and treated (and whether there were any differences in the way these were identified), patient-level outcomes, and for the training 'change of knowledge, attitudes and practice' (KAP) and satisfaction questionnaires.<br> – Key informant interviews and/or focus group discussions with key stakeholders such as people who received the training and/or those who delivered or received the interventions; cost data will be collected from programme managers. | The exact evaluation plan will be developed and finalised following phases 1 and 2, but will include evaluation of the care package in regards to coverage (ie, the degree to which affected persons in the selected districts actually received the care package), as well as implementation outcomes (including acceptability), clinical (physical health, mental health and psychosocial), economic and social outcomes, similar to the evaluation during the pilot study during phase 2 (see row above and table 2). Evaluation data will be collected at baseline, and at 3-month and 12-month follow-up. |

FAST, Fast Alcohol Screening Test; FMOH, Federal Ministry of Health; LF, lymphatic filariasis; MMDP, morbidity management and disability prevention; NTD, neglected tropical disease.

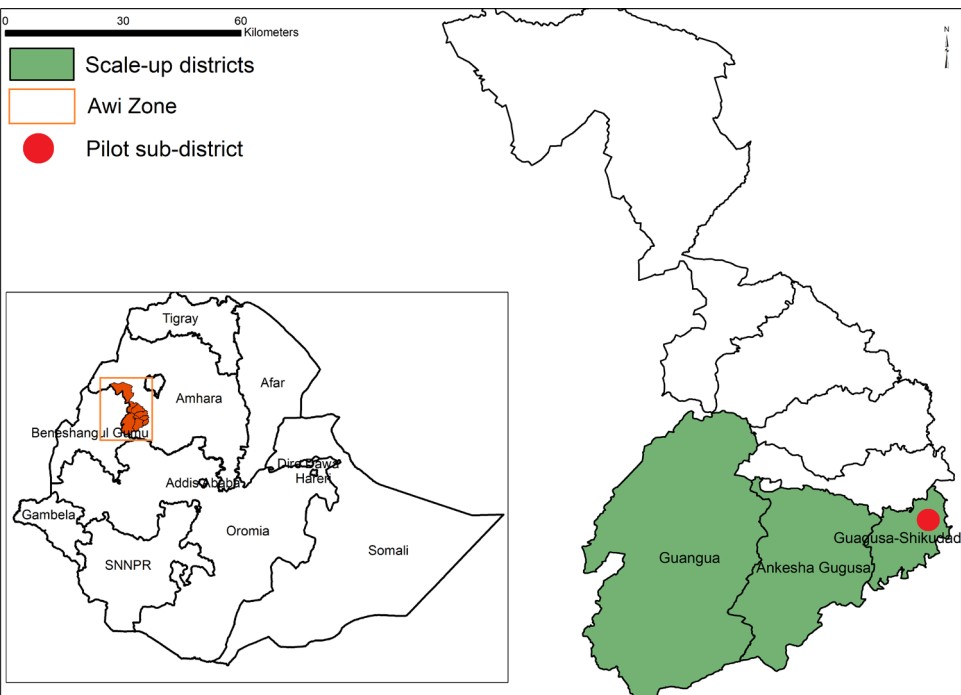

**Figure 2** Implementation districts for the Excellence in Disability Prevention Integrated across NTD (EnDPoINT) study. NTD, neglected tropical disease.

during phase 2; and three whole districts for the scale-up during phase 3 (Guagusa Shikudad, Ankesha Guagusa and Guangua). The districts were agreed between the EnDPoINT research team and the Ethiopian FMOH based on: coendemicity of podoconiosis, LF and leprosy; any previous or ongoing work in the districts; and accessibility. The population of Awi zone is 982 942 (with each district/*woreda* having populations of between around 8000 and 31 500), according to the latest Ethiopian Census in 2007.[48] As for most of Ethiopia, the zone is largely a rural area (around 87.5% of the population[48]); Agew-Awi (59.8%) and Amhara (38.4%) are the two main ethnic groups, and the two most commonly spoken first languages are Amarigna (53.4%) and Agew-Awigna (45.0%). The large majority of the population (94.4%) is Ethiopian Orthodox Christian.[48]

A wide range of stakeholders are involved in EnDPoINT. For the qualitative research activities in all three phases of the study, situational analysis in phase 1, and the training components in phases 2 and 3, purposive sampling techniques are being used to identify, approach and recruit key stakeholders based on their role and position. Members of the EnDPoINT Consortium initially contact key stakeholders to ask them to participate in the particular component of the study. Snowballing techniques may also be employed, whereby each of the key stakeholders identified is asked whether they are aware of any other key stakeholders who they think should be included.

For the qualitative work, such as the workshops, key informant interviews and focus groups, participants include members of the EnDPoINT study consortium who have expertise in the provision and/or receipt of care and the context in Ethiopia, current NTD and/

or mental health programme implementers (including non-governmental organisations), personnel in the NTD Department of the Ethiopian FMOH, policy-makers, health service planners, health managers and decision-makers, facility-based health workers (eg, primary health-care staff, health officers, nurses), community health workers (CHWs), community leaders, traditional and religious leaders, members of service user organisations, as well as affected persons themselves and their families.

For the situational analysis in phase 1, key health officials and service managers are approached based on purposive sampling methods if necessary, though most of the information collected for this is being based on secondary data that are available in the public domain.

During the pilot study and scale-up of the care package in phases 2 and 3, key stakeholders are purposively selected to receive training to implement the various components (interventions) of the care package; this includes staff at the healthcare organisation level, zonal and district level health office staff, psychiatric nurses, senior healthcare workers, pharmacy staff, health centre staff (health officers and nurses), CHWs/HEWs, and members of the community (ie, community leads as facilitators of community conversations (CC)).

Affected people, that is, those with lower limb lymphoedema caused by podoconiosis, LF and leprosy, as well as their families and communities, are being included throughout the study; these are people identified as requiring care for their lymphoedema and/or comorbid mental ill-health. Patients are identified and recruited into the study based on their case identification within the EnDPoINT programme, that is, all adult patients (ie, patients over 18 years of age) within the subdistrict in

phase 2 and the three districts during phase 3 who are identified as having either lymphoedema due to podoconiosis, LF or leprosy will be invited into the study (a few of whom will be selected as case studies). Apart from being beneficiaries of some interventions, affected persons will also potentially contribute to the delivery of some of the training packages and/or interventions, for example, by contributing to training sessions in order to provide a 'lived testimony' of living with the disease, facilitating self-help groups, and being actively involved in awareness-raising and stigma reduction activities such as CC; this is because of growing evidence that social contact elements within interventions, whereby people come into contact with affected persons and hear their stories, are most effective in reducing stigma and discrimination, and can contribute to increased service uptake and provision.

Only adults are being included in the study, that is, participants are all at least 18 years of age. All components of the EnDPoINT study are being conducted in either Amharic or English, depending on which language is most suitable to the particular participant group. Where necessary, all study instruments and materials, including participant information sheets and consent forms, are translated into Amharic, which is the official language of Ethiopia and can be spoken and understood by the majority of people in Ethiopia, including in Awi zone.

### Sample size justification

For the qualitative research activities, such as key informant interviews and focus group discussions, the sample size is guided by the number of key stakeholders who are identified to take part. A data saturation approach is being used (ie, data collection will continue until sufficient information has been obtained or where further data collection fails to generate new/additional themes), which is usual and appropriate for qualitative data collection techniques. For the workshops and training components, the sample size will be established based on the number of people who are identified to be suitable and available. However, the sample size during any workshops and focus groups will not exceed 16 participants, to ensure that all participants have the opportunity to speak.

During phase 2, all health staff in the selected subdistrict are trained. During phase 3, in the three districts, five health workers from each health centre will be trained in the 13 health centres identified through the situational analysis in phase 1 and they will cascade the training to the remaining 15 staff at each health centre; a total of 150 HEWs in the three districts will be trained, which is around 10 health professionals across the three districts (there are 4–5 health facilities per district). Similarly, all adult patients who are identified within the single subdistrict during phase 2 and within the three whole districts during phase 3 as having either lymphoedema caused by podoconiosis, LF or leprosy will receive the care interventions included in the care package (or will at least be offered these); we are expecting to identify around 1500–1600 people with lymphoedema due

to podoconiosis, leprosy or LF in total across the three districts during phase 3 based on previous surveys that have been conducted (we do not have numbers available for LF and leprosy).

### Key interventions within the care package

The EnDPoINT care package is organised according to three levels of the healthcare system: healthcare organisation/coordination, healthcare facility and community level. This may include the following interventions, though these will be finalised only after completion of phase 2 due to the iterative nature of the study:

► Healthcare organisation/coordination:
  – High-level awareness-raising and mobilisation: participatory workshop for zonal and district health bureau officials, and evidence generation for best practice.
  – Programme management, that is, working in partnership with key healthcare organisation staff to ensure the necessary structures and budget for the delivery of the care package.
  – Capacity-building, that is, 'Training of Trainers' (ToT) for healthcare coordination staff about lymphoedema MMDP and mental health care provision, and supportive supervision, mentoring and coaching.

► Health facility:
  – Capacity building for facility-based health centre staff: training on lymphoedema MMDP and mental health care provision, and training on supply chain management for those staff who manage MMDP supplies.
  – Awareness-raising and stigma reduction: (1) participatory workshop for facility-based staff; (2) posters in health centres, including waiting rooms; (3) health education sessions by health facility staff for attendees of health facilities.
  – Case detection, assessment and treatment initiation by health centre nurses and health officers, including training of affected persons in self-care, counselling and coping skills acquisition for affected persons, and mental healthcare.
  – Clinical mentoring for facility-based health workers.
  – Supportive supervision for facility-based health workers and CHWs.

► Community:
  – Capacity-building for CHWs: (1) training on lymphoedema MMDP and mental healthcare provision; (2) CC facilitator training.
  – Community awareness-raising and stigma reduction: (1) awareness-raising workshops for members of the community by CHWs; (2) CC groups; (3) information dissemination in the community by CHWs (information leaflets, posters in health posts, mass media).
  – Active case detection and referral by CHWs.
  – Patient follow-up visits by CHWs.

– Community-based rehabilitation, including home visits, meetings, workshops, patient associations and self-help groups, family support groups, and community mobilisation.

– Supportive supervision for CHWs.

In addition, cutting across the three levels of the healthcare system, there may also be a community advisory group of a wide range of stakeholders across the three levels. It is important to ensure high quality of the healthcare providers so that they are able to combine physical health, mental health and psychosocial care.

## Procedures

See table 1 for an overview of the research activities/methods conducted during each of the three phases of the EnDPoINT study.

### Development of care package during phase 1

During phase 1, a document review of grey literature and a separate systematic review are conducted to inform and guide the development of the holistic care package and strategies for its integration into the routine healthcare delivery system in Ethiopia. To contribute further to this, a situational analysis/resource assessment is carried out to collect cross-sectional baseline data on factors relevant to the development and integration of the care package in Awi zone in Ethiopia, as well as any resources that are available for this, and potential risk factors. Three ToC workshops are conducted to map out the care package's ToC in terms of defining its desired outcomes, indicators, interventions and measurement of the outcomes, which are represented graphically on a ToC map along with an accompanying narrative describing the key pathways and assumptions, as well as to encourage stakeholder buy-in to the study.[45] Key informant interviews and focus groups are then conducted, to test the feasibility, acceptability and appropriateness of the draft care package, as well as to assess key aspects of the ToC, for example, some of the key assumptions identified during the workshops.[45] Participants of this qualitative work and the ToC workshops include affected persons, caregivers and various health providers. The care package and its associated training materials are discussed and finalised following a workshop with members of the EnDPoINT research team and FMOH in Ethiopia. A separate manuscript will be prepared detailing the phase 1 activities further.

A large focus of the research content during all of the phase 1 research activities outlined above is the mental health and psychosocial components of the care package, including the stigma reduction elements, as well as how to integrate care across the three diseases and into the routine healthcare system, as this is where there are gaps in knowledge within the Ethiopian context. In line with this, based on the research activities during phase 1, already-existing MMDP guidelines for podoconiosis and LF are adapted as part of the project to include leprosy and mental health components, to inform the training materials.

### Piloting of care package during phase 2

During the pilot study in one subdistrict in Awi zone, the main training elements and interventions of the care package are tested through a mixed-methods design, to establish their adoption, feasibility, acceptability, fidelity, potential effectiveness, their readiness for scale up, costs of the interventions, and the suitability of the training and training materials. This involves implementing the interventions, including their training components, as outlined in the section on the draft care package above, and evaluating them. During the pilot study, the interventions covered by the training are implemented for 3 months.

Quantitative evaluations of the interventions and associated trainings are conducted at baseline and 3-month follow-up through surveys with those affected people who are receiving the treatment interventions within the care package, to measure changes in patient outcomes in relation to physical health, mental health/well-being, quality of life, stigma and discrimination (including social support), disability and healthcare use and costs; surveys with members of the community within which the affected persons live, to assess changes in stigma and 'change of knowledge, attitudes and practice' (KAP); as well as an evaluation of the training through assessment of KAP of health workers receiving the training and their satisfaction with the training. In addition, qualitative assessments are carried out through observation of the interventions being implemented; and key informant interviews and focus group discussions with those stakeholders who received the training and delivered the interventions, as well as the recipients of the interventions (ie, people affected by podoconiosis, LF and leprosy). Table 2 shows the outcomes that are assessed in phase 2 together with how these are measured; see also right-hand column in table 1 for the outcome measurements used. All measurement tools are administered by data collectors rather than being completed by participants themselves, meaning that illiterate participants are not disadvantaged.

The care package, as well as the training materials, may be revised further based on the results of the pilot study (success criteria will be established a priori), and there may be further pilot testing if major challenges are encountered that will need to be addressed further, before being scaled up in three whole districts.

### Scale-up of care package during phase 3

The scale-up and evaluation of the care package during phase 3 of the EnDPoINT study will involve implementation of all interventions at all three levels of the health system (healthcare organisation/coordination, health facility and community level) in three districts in Awi zone. While the exact interventions that will be included in the care package for scale-up will depend on the findings of phases 1 and 2, the general potential structure of the training for the various interventions included in the care package is depicted in figure 3.

**Table 2** Outcomes and their measures during phases 2 and 3 of the Excellence in Disability Prevention Integrated across NTDs (EnDPoINT) study

| Type of outcomes | Specific outcome | Outcome measures |
|---|---|---|
| Implementation outcomes | Adoption | Qualitatively (KIIs/FGDs, observation) |
| | Feasibility | Qualitatively (KIIs/FGDs, observation); cost of training/ interventions; number of cases identified, assessed and treated |
| | Acceptability (by providers and affected persons) | Qualitatively (KIIs/FGDs, observation) |
| | Appropriateness | Qualitatively (KIIs/FGDs, observation) |
| | Fidelity | Qualitatively (KIIs/FGDs, observation) |
| | Readiness for scale-up | Qualitatively (KIIs/FGDs, observation) |
| | Economic characteristics of study participants | Purposely designed questionnaire |
| | Resource use associated with delivering the care package | Project financial records and interviews with project manager(s) |
| Effectiveness (patient level) | MMDP assessment | Swelling circumference (physical measurements); frequency of acute attacks (patient self-report); stage/ grade of affected limb (physical assessment); signs of infection (physical assessment); wounds (physical assessment); nodules (physical assessment) |
| | Depression | Patient Health Questionnaire 9 |
| | Suicidal ideation and action | CIDI questions |
| | Alcohol use | Fast Alcohol Screening Test |
| | Quality of life | Dermatology Life Quality Index |
| | Disability | WHO Disability Assessment Schedule 2.0 |
| | Internalised stigma | Internalised stigma related to lymphoedema |
| | Discrimination | Discrimination and Stigma Scale 12 |
| | Social support | Social Support Scale (OSLO-3) |
| | Happiness | Happiness index |
| | Use of primary care/cost | Purposely designed questionnaire |
| | Use of hospital care/cost | Purposely designed questionnaire |
| | Use of medication/cost | Purposely designed questionnaire |
| | Personal expenses | Purposely designed questionnaire |
| | Days off work due to illness/cost | Purposely designed questionnaire |
| Effectiveness (community level) | Coverage | Number of cases identified, assessed and treated; proportion of cases detected who are then treated; number of affected persons reached with MMDP supplies; number of affected persons who have received mental healthcare; contact coverage |
| | KAP lymphoedema | Purposely designed questionnaire |
| | Social distance | Social Distance Scale (SDS) |
| Effectiveness (facility level) | KAP lymphoedema | Purposely designed questionnaire |
| | KAP mental health | Purposely designed questionnaire |
| Other outcomes | Suitability of training and training materials | Qualitatively (KIIs/FGDs, observation) |
| | Satisfaction with training | Purposely designed questionnaire; qualitatively (KIIs/ FGDs, observation) |

FGDs, focus group discussions; KAP, knowledge, attitudes and practice; KIIs, key informant interviews; MMDP, morbidity management and disability prevention.

The scale-up of the care package interventions during phase 3 will be implemented, followed up and evaluated over a period of 1 year. Affected persons will be followed up every month for the first 3 months and then at 12 months. A mixed-method approach will be used for the evaluation of the care package. A pre–post design will be employed for quantitative evaluation of the care package, with data collected at baseline (ie, before training or any of the intervention components commence), and at 3-month and 12-month follow-up; data will be collected

**Figure 3** Potential training structure within the Excellence in Disability Prevention Integrated across NTD (EnDPoINT) study. MMDP, morbidity management and disability prevention.

on coverage, as well as implementation, clinical (physical health, mental health and psychosocial), economic (including total costs of the care package) and social outcomes (see table 1). The questionnaires for evaluation are the same ones used during the phase 2 piloting, though these may be refined and improved based on the piloting. A detailed monitoring and evaluation plan for phase 3 will be developed based on the results in phase 2.

For the qualitative evaluation of the care package during phase 3, key informant interviews and focus group discussions will be carried out with a wide range of stakeholders who have been involved in the implementation of the care package (eg, affected persons and their families, health facility staff, community health workers, health facility management staff, district-level health staff, etc); these will be carried out at two time points, once towards the beginning of the scale-up period and then again towards the end of the scale-up period. The topic guides and interview questions for the key informant interviews and focus group discussions will be developed once phase 2 has been completed.

### Outcomes to be measured

Tables 1 and 2 detail the outcomes measured during EnDPoINT, as well as the time points and methods that will be used to measure these.

### Data analysis plan

Qualitative data (such as from the key informant interviews and focus group discussions) will be analysed using thematic analysis with the assistance of a qualitative software package (NVivo). Validity (dependability) of data will be assessed through the identification of the initial codes by an experienced qualitative researcher and agreeing on the framework in which the data will be presented, presentation of initial findings to experienced academics to assess the plausibility, and including single counting of identified events/phenomena. Validity assurance will also involve checking of alignment of research questions, interview guides, sampling procedures and so on. For quantitative data collected during phases 2 and 3, descriptive analyses (such as simple counts and frequencies or means and SD) will be used, as well as the % change before and after the implementation of the interventions, together with 95% CIs). A detailed data analysis plan will be produced for the evaluation of the care package during phases 2 and 3. The results from the various research activities throughout the study will be triangulated to develop and finalise the holistic care package and strategies for its integration into the routine healthcare delivery system in Awi zone in Ethiopia.

### Patient and public involvement

Affected persons and their communities are being fully engaged in the research process throughout the entire EnDPoINT project. EnDPoINT is being supported by a consortium, which includes patient representatives alongside researchers, health professionals, FMOH staff and other stakeholders, whose remit is to provide input, guidance and advice throughout the programme of research. Affected persons and community members are partnering with the EnDPoINT study team for the design of the study and care package. This includes the development and implementation of the interventions; the development of materials, such as training materials; the sharing and identification of findings; and the identification of appropriate channels and venues for the dissemination of results. Affected people and their families and communities are included within the workshops and qualitative aspects of the work, such as the key informant interviews and focus group discussions. The situational analysis ensures that the interventions fit in with the local context and build on the strengths and assets of the community. Where possible, members of the local community are being employed to collect data, to facilitate capacity building of community members. The EnDPoINT Consortium will contribute to various outputs for dissemination. Affected persons and community members are therefore being involved in the entire study process, both as coresearchers and as participants of the study.

### Economic analysis

We aim to cost the holistic package of physical health, mental health and psychosocial care interventions for people with lower limb lymphoedema caused by podoconiosis, LF or leprosy, and the integration of this package into the routine healthcare delivery system (phases 2 and 3), thereby ensuring that the care package represents value for money. Intervention activities will be costed at three levels:

1. Healthcare organisation/coordination level: awareness-raising activities, programme management and training healthcare organisation staff.
2. Healthcare facility level: awareness-raising activities, case detection, treatment, staff training in health education for patient physical health self-care, mental health and psychosocial support.
3. Community level: awareness-raising activities, training and supervision of CHWs, active case detection by CHWs, providing community-based healthcare and rehabilitation.

The main cost components will be identified in the pilot study (phase 2); cost data will be collected during the scale-up of the care package intervention (phase 3). A microcosting approach will include a bottom-up construction of costs associated with setting up and delivering the package. The costs of training healthcare professionals will include: time spent in preparing and attending the training sessions, room rental, training materials, travel expenses and administration costs. The costs of delivering the healthcare package will include: costs of lymphoedema treatment supplies (eg, soap, ointment, bandages, gloves, antiseptics), infrastructure costs and additional staff salaries/time. Costs will be collected from financial records and interviews with programme managers. We will also collect data on the socioeconomic characteristics of study participants, housing, economic activities, use of healthcare services and out-of-pocket expenses related to the lower limb lymphoedema. Direct costs will cover contacts with healthcare professionals, pharmacists and traditional healers, medication and hospitalisations with respect to the conditions. Indirect costs will include additional costs to the individuals and their families (eg, travel to hospital, accommodation, food and borrowing money from family/relatives/community). Economic data will be collected using a purpose-designed questionnaire.

## ETHICS AND DISSEMINATION
### Ethics
Ethical approval has already been obtained from the Brighton and Sussex Medical School Research Governance and Ethics Committee in the UK, as well as the Institutional Review Board of the College of Health Sciences at Addis Ababa University in Ethiopia.

All people who are participating in the research (all over 18 years old) are given a participant information sheet (outlining the study's research aims, procedures, and potential risks and benefits) and are required to give their written consent to take part (or verbal consent with a witness declaration by the interviewer where a person is illiterate); if a person does not consent to take part, that person is excluded. However, those people who are involved purely in the implementation of the care package, for example, those who are receiving training on self-care for lymphoedema or who are taking part in a stakeholder meeting, are not required to give their informed consent, though they are not obliged to be involved. All participants are able to withdraw from the study at any time without having to give a reason or any detrimental consequences, for instance in regard to their care, treatment or employment, until data have been aggregated.

Confidentiality and anonymity of data are ensured throughout the study process, from data collection to data storage, analysis and publication. Structured, multilevel precautions are taken to safeguard the confidential nature of the information gathered, and to ensure the anonymity of participants, for example, by using personal identifiers instead of names, with identifying data kept separately from the identifying codes used on the data collection sheets and databases. Only anonymised data will be used for data analyses. All data are being stored in a secure OneDrive folder, which can only be accessed by members of the team for whom access to the data are essential.

### Dissemination
An EnDPoINT publication plan has already been developed, which includes several planned articles for publication in scientific journals. It is likely that the results of the study will also be shared through various other forums, including to the larger scientific community through conferences, but also to policy-makers and healthcare staff, and other participants of the study. This will likely be in the form of a workshop at the end of the study and/or through the distribution of a summary document and/or policy brief(s) with an overview of the main results of the study.

**Author affiliations**
[1]Centre for Global Health Research, Brighton and Sussex Medical School, Brighton, UK
[2]Center for Innovative Drug Development and Therapeutic Trials for Africa (CDT-Africa), Addis Ababa University, Addis Ababa, Ethiopia
[3]School of Public Health, College of Health Sciences, Addis Ababa University, Addis Ababa, Ethiopia
[4]Department of Primary Care and Public Health, Brighton and Sussex Medical School, Brighton, UK
[5]Centre for Neglected Tropical Diseases, Department of Tropical Disease Biology, Liverpool School of Tropical Medicine, Liverpool, UK
[6]Health and Development Cluster, Institute of Development Studies, University of Sussex, Brighton, UK
[7]Faculty of Medicine, School of Public Health, Imperial College London, London, UK
[8]College of Humanities, Language Studies, Journalism and Communication, Addis Ababa University, Addis Ababa, Ethiopia
[9]Neglected Tropical Diseases, Disease Prevention and Control Directorate, Federal Ministry of Health, Addis Ababa, Ethiopia

**Acknowledgements** The authors thank the other members of the EnDPoINT Research Consortium for their input into the study, as follows: Tsige Amberbir, Tanny Hagens, Mossie Tamiru, Tadesse Tesfaye, Seifu Tirfie, and Abebayehu Tora. They also thank Clare Callow for her project management of the NIHR grant through which EnDPoINT is funded, as well as Tesfaye Asefa, Bethelhem Fekadu, Grit Gansch, Samrawit Ketema for their support with the administrative aspects of the study. Our sincerest thanks to all participants of the EnDPoINT study.

**Contributors** All authors contributed to the manuscript and have accepted the final version. MS led the writing of the manuscript. GD and AF are the principal investigators of the EnDPoINT study. OA, AM, AT, MK, SAB and NH are part of the core EnDPoINT research team. OA, AM, AT and MK are responsible for the

implementation of the study in Ethiopia. SAB leads the statistical elements of the study. NH leads the economic aspects. KD produced two of the figures in the manuscript, and is a member of the EnDPoINT Research Consortium, as are LAK-H, HM, HBT, HB, DH and NN. All members of the EnDPoINT Research Consortium have actively contributed to the inception of the study and continuous deliberations since, in relation to conceptual and methodological approaches and implementation. AF and GD contributed equally as joint last authors.

**Funding** This research was commissioned by the National Institute for Health Research Global Health Research Unit on NTDs at BSMS using Official Development Assistance funding. KD is supported by the Wellcome Trust as part of his International Intermediate Fellowship [grant number 201900].

**Disclaimer** The views expressed in this publication are those of the author(s) and not necessarily those of the NIHR or the Department of Health and Social Care.

**Map disclaimer** The depiction of boundaries on this map does not imply the expression of any opinion whatsoever on the part of BMJ (or any member of its group) concerning the legal status of any country, territory, jurisdiction or area or of its authorities. This map is provided without any warranty of any kind, either express or implied.

**Competing interests** None declared.

**Patient and public involvement** Patients and/or the public were involved in the design, or conduct, or reporting, or dissemination plans of this research. Refer to the Methods section for further details.

**Patient consent for publication** Not required.

**Provenance and peer review** Not commissioned; externally peer reviewed.

**ORCID iDs**
Maya Semrau http://orcid.org/0000-0003-0366-1093
Abebaw Fekadu http://orcid.org/0000-0003-2219-0952

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
