## [Reviewer comments · BMJ Open]

ARTICLE DETAILS

TITLE (PROVISIONAL)	EnDPoINT – Protocol for an implementation research study to integrate a holistic package of physical health, mental health and psychosocial care for podoconiosis, lymphatic filariasis and leprosy into routine health services in Ethiopia
AUTHORS	Semrau, Maya; Ahmed, Oumer; Deribe, Kebede; Mengiste, Asrat; Tesfaye, Abraham; Kife, Mersha; Bremner, Stephen; Hounsoume, Natalia; Kelly-Hope, Louise; MacGregor, Hayley; Taddese, Henock; Banteyerga, Hailom; HaileMariam, Damen; Negussu, Nebiyu; Fekadu, Abebaw; Davey, Gail

VERSION 1 – REVIEW

REVIEWER	Shahed Hossain BRAC James P. Grant School of Public Health, BRAC University, Bangladesh
REVIEW RETURNED	19-Mar-2020

GENERAL COMMENTS	This is a well written protocol covering all standard components of implementation research. Few comments to strengthen the protocol: 1. Consider including possible challenges or risks (if risk analysis is done) to implement the protocol (a log frame analysis with SWOT analysis)2. Do authors like to consider direct or indirect cost incurred by the patients?3. A very strong process documentation and monitoring mechanism is essential for this large protocol to implement through phases of implementation. what additional measure the authors would likely to take to strengthen the public sector health systems (e.g., quality HIS, local level planning etc.)? Thanks.
--

REVIEWER	Giulia D'Odorico LSHTM - The London Institute of Hygiene and Tropical Medecine – UK
REVIEW RETURNED	07-May-2020

GENERAL COMMENTS	Even if the protocol is well structured and the main objective and results are clearly defined, the article will need some major revision to allow the study to be repeated. Actually much information which could help to understand the applicability of the project is missing. Authours could better explain how they will design and develop the comprehensive package of holistic physical health, mental health and psychosocial care. In different parts of the protocol (for instance in the abstract and the study design) the authours make reference to the development of the care package and this objective is quite clear
---

in Table 1. However, it is not clear how the care package will be designed. Authors should describe what is already existing in terms of care package and what are the main gaps, what are the main research topics to be addressed during the design, the methodologies, the participants, etc. This is particularly missing for the social component of the project (mental health and psychosocial care, stigma reduction) but also for the clinical component.

Some research questions are not logically interconnected between Phase 1 and 2. For instance some research questions of Phase 2 ('What are the critical contextual factors (including drivers and barriers) that influence the process of integration of the holistic care package into government-run health services and across the three diseases? What are the key features of the intervention that influence the manner of integration into the health care system and across the three diseases? Who are the key health system actors that have a stake in the integration of the care package into the government-run health services, and what coordination and capacity building need exist?') should already be answered during phase 1 in order to be able 'to inform the development of the core package of care and strategies for its integration into the routine health care delivery system, as well as integration of MMDP across the three diseases' as foreseen (line 43-53 p. 6). ToC/interviews and FGDs can be used to explore and understand the factors that influence integration of the care package, the roles of the key actors, etc. Then, the collected data can be used to design the care package and not just to assess the draft care package and training materials. How and when these care package and training materials have been developed? This is not clear in the protocol.

A detailed description of the research contents should appear in the description of phase 1. What are the main research topics? Why we want to work on them? What has been done up to know and what are the main gaps? A reader can have the impression that research contents are taken for granted. They should be discussed with stakeholders during Phase 1 and adapted to the context. This would make the research project more meaningful for the stakeholders who will implement it and benefit from it.

Evaluation methodologies for the care package described in the protocol need to be refined. Only in the Table 1 authors describe the evaluation methodologies to be used for the assessment of the effectiveness of the care package. This information should be included also in the part dedicated to the piloting of the care package in phase 2 in the document. Moreover, in Table 1 we can find only a list of assessment tools, while objectives and targets are not well defined.

The involvement of patients within the research project should be better defined. Although authors make clear that patients will have an active role into the development and implementation of the project (10-19 p. 13), it is not clear how they will make this feasible. Authors can describe better how a patient-centered approach can help the project to be better grounded in the communities; what is a Consortium; how members are recruited (selection criteria); when and how patients will be involved into the research process. For instance, how patients are involved in phase 1 to contribute to the design of the care package? Which part of the care package, etc.? How they will be involved into trainings (6-12 p. 10) ? Why patients are involved just to do ternoignages during trainings while they can

	have a more active role? For instance, if it is a psycho-social training, they can actually be trainers, for instance describing the challenges patients may face within their care pathway and how the health system/community/family, etc. can support them. They could be trained/act as peer educators, instead of just participating into contact events. The project foresees also awareness-raising and stigma reduction activities (line 12-15 p. 9). During the design of the care package, some other activities (interviews, focus groups, etc.) can be foreseen to study meanings and experiences around stigma, so to make sure that awareness-raising and stigma reduction activities are based on this and adapted to the needs of those who are stigmatized. Authors should better describe why the project wants to address stigma (is it a barrier to access to health? Or they want to focus on the psycho-social consequences of stigma on affected people, their family, entourage, etc.) and who they want to target (communities? health centres? individuals - patients, caregivers, family members, etc.). This information could help the authors to develop better strategies to reduce stigma and appropriate tools. Moreover, HP stigma-related activities may be conducted by former/current patients instead of health staff: this could be an empowering instrument for patients, instead of reducing them only to activities such as 'tmoignages'. Some patient counselling guidelines among the HP materials (line 17 p. 9) are quoted in the protocol. Are they already existing or they should be developed? If yes, will the project contribute to this and how? This could be a very interesting contribution to the NTDs-related research and to national curricula of health workers. Outcomes could be better described and to be more specific. Authors describe how they will measure outcomes/evaluate the care package but they do not explicit them. Is the increased number of patients identified and linked to care an outcome? The decreasing of loss-to-follow up? An improvement of the mental wellbeing of patients? A decrease in reported and experienced stigma in the health centre/in the community? Increased capacity of HWs to reliably diagnose and manage the diseases. This is maybe due to the fact that objectives for each component of the care package may not be very clearly defined. There are also some other small details which could improve the completeness of the protocol. In Figure 3 authors refer to training members of the community. Why we involve them? To improve early case detection? Reduce stigma? Who are them? Opinion leaders? Which are the selection criteria? This should be better clarified. Concerning data management (line 44 p. 10) I would add some more details about where all the electronic fieldnotes, interview/FGDs audio-recordings and transcripts are stored. Authors could include the potential direct/indirect costs of integrated care for patients and their families into the economic analysis (23-50 p. 11), while not focusing exclusively on the care package costs. Some more references may be added with reference to integrated care models for NTDs even from other countries, such as Prochazka 2020 in Liberia, Barogui 2020 and Koffi 2020 in Ivory Coast. A timetable may be useful to better understand the different phases of the project.
--	---

REVIEWER	Peter Steinmann Swiss Tropical and Public Health Institute, Switzerland
-----------------	--

REVIEW RETURNED	03-Jun-2020
-------------

GENERAL COMMENTS	Semrau and co-authors present the protocol for a comprehensive lymphedema prevention and care package integrated into routine health care and targeting three debilitating and stigmatizing neglected tropical diseases. The protocol is well written and the program carefully developed. Below, a couple of comments that might help the authors to further specify certain aspects of the proposed implementation research study.  - The protocol appears to suggest that lower limb lymphedema were the main consequence of LF, podoconiosis and leprosy. However, particularly in the case of leprosy there are numerous other health problems associated with more severe cases, particularly those with disability grade 2. It is unclear how these will be considered to make the care package truly holistic. - Expansion to the whole zone is proposed as the endpoint for the project. To secure knowledge and maximise the impact it seems important to already plan for the time beyond the lifetime of the project (sustainability of intervention) and how findings might be integrated into national policy (policy briefs are mentioned) and practice (roll-out). - The project aim is described as “facilitate effective access”. The demonstration of feasibility, acceptability and perception might be emphasized more strongly. - It seems appropriate to specifically mention acceptability and perception assessments also in phase 3 (roll-out) - Procedures to accommodate the specific needs of illiterate participants are mentioned for informed consent but not for the study data collection. It is mentioned that English study instruments and materials will be translated into Amharic, but not how they will be made accessible for illiterates. Also, it is not mentioned whether all potential participants speak Amharic as also other languages seem to be common in the region. - The sample size section would benefit from some indications on likely sample sizes that will be targeted/can be expected.
---

VERSION 1 – AUTHOR RESPONSE

Reviewer 1

This is a well written protocol covering all standard components of implementation research.

Response: We thank the reviewer for this kind comment.

Few comments to strengthen the protocol:

1. *Consider including possible challenges or risks (if risk analysis is done) to implement the protocol (a log frame analysis with SWOT analysis)*

Response: Thank you for this comment. We agree that it is important to assess challenges and risks. We did this through a ‘Theory of Change’ approach and situational analysis, to map risks (assumptions) and to understand how we might deal with them. Some further details on these activities have been added to the manuscript (see page 11 in the marked version of the manuscript).

2. *Do authors like to consider direct or indirect cost incurred by the patients?*

Response: Thank you for this comment. We have added some details on direct and indirect costs in the manuscript, in the 'Economic analysis' section (see page 14).

3. *A very strong process documentation and monitoring mechanism is essential for this large protocol to implement through phases of implementation. What additional measure the authors would likely to take to strengthen the public sector health systems (e.g., quality HIS, local level planning etc.)?*

Response: The project will contribute to health system strengthening through lymphoedema service delivery, training of the health workforce on management of lymphoedema, and integration of key indicators for lymphoedema management into the health management information system. The project will also identify medical supplies important for morbidity management to be included in the list of essential medicines and supplies list. Finally, the project will work with district-level health planners over resource allocation for lymphoedema management. Key measures to monitor this will include adherence to the intervention, markers of success, acceptability, data quality etc.

Reviewer 2

1. *Even if the protocol is well structured and the main objective and results are clearly defined, the article will need some major revision to allow the study to be repeated. Actually much information which could help to understand the applicability of the project is missing.*

Response: We thank the reviewer for these comments. We have now included more details about the project in line with the reviewer's comments below, which we believe has greatly strengthened the manuscript.

2. *Authors could better explain how they will design and develop the comprehensive package of holistic physical health, mental health and psychosocial care. In different parts of the protocol (for instance in the abstract and the study design) the authors make reference to the development of the care package and this objective is quite clear in Table 1. However, it is not clear how the care package will be designed. Authors should describe what is already existing in terms of care package and what are the main gaps, what are the main research topics to be addressed during the design, the methodologies, the participants, etc. This is particularly missing for the social component of the project (mental health and psychosocial care, stigma reduction) but also for the clinical component.*

Response: Thank you for this helpful comment. We agree that some further detail in regards to the development of the care package during Phase 1 of the study could be helpful. We have now added further information on this in the manuscript, both in regards to gaps and elements that were added to the care package during the project (see 'Study design' section on page 7 in the marked version of the manuscript), as well as procedures/methods for how the care package was developed (see 'Procedures' section on page 11). A separate manuscript is being prepared on the development of the care package, which will provide more in-depth information about all of the activities conducted within Phase 1 – we have now pointed to this further manuscript in our manuscript (see page 11).

3. *Some research questions are not logically interconnected between Phase 1 and 2. For instance some research questions of Phase 2 ('What are the critical contextual factors (including drivers and barriers) that influence the process of integration of the holistic care package into government-run health services and across the three diseases? What are the key features of the intervention that influence the manner of integration into the health care system and across the three diseases?')*

Who are the key health system actors that have a stake in the integration of the care package into the government-run health services, and what coordination and capacity building needs exist?') should already be answered during phase 1 in order to be able 'to inform the development of the core package of care and strategies for its integration into the routine health care delivery system, as well as integration of MMDP across the three diseases' as foreseen (line 43-53 p. 6). ToC/interviews and FGDs can be used to explore and understand the factors that influence integration of the care package, the roles of the key actors, etc. Then, the collected data can be used to design the care package and not just to assess the draft care package and training materials. How and when these care package and training materials have been developed? This is not clear in the protocol.

Response: Thank you for highlighting that not all of the research questions were logically connected between Phase 1 and Phase 2. We agree that some of the research questions were misplaced. We have now rectified this by adding some of the research questions from Phase 2 under Phase 1 (which however are sustained in Phase 2) (see Table 1, pages 20-22), as well as within the abstract (page 2) and section on 'Study design' (page 7). In regards to how and when the care package and training materials were developed, we hope that this is now clearer in the manuscript through the added section on 'Procedures' for Phase 1 (see page 11), in response to your comment 2 above.

- 4. A detailed description of the research contents should appear in the description of phase 1. What are the main research topics? Why we want to work on them? What has been done up to know and what are the main gaps? A reader can have the impression that research contents are taken for granted. They should be discussed with stakeholders during Phase 1 and adapted to the context. This would make the research project more meaningful for the stakeholders who will implement it and benefit from it.*

Response: We have added some information to the manuscript in regard to the focus of the research contents during Phase 1 (see 'Procedures' section on page 11).

- 5. Evaluation methodologies for the care package described in the protocol need to be refined. Only in the Table 1 authors describe the evaluation methodologies to be used for the assessment of the effectiveness of the care package. This information should be included also in the part dedicated to the piloting of the care package in phase 2 in the document. Moreover, in Table 1 we can find only a list of assessment tools, while objectives and targets are not well defined.*

Response: Thank you for this comment. We have now included information on evaluation methodologies used during Phases 2 and 3 of the study in the text of the manuscript (see 'Procedures' section on pages 11-12 and 12-13), and for Phase 3 in Table 1 (pages 22-23). Furthermore, a table has been added (Table 2; see pages 24-25), which lists the outcomes evaluated in Phases 2 and 3 together with the tools/methods that are being used to measure them.

- 6. The involvement of patients within the research project should be better defined. Although authors make clear that patients will have an active role into the development and implementation of the project (10-19 p. 13), it is not clear how they will make this feasible. Authors can describe better how a patient-centered approach can help the project to be better grounded in the communities; what is a Consortium; how members are recruited (selection criteria); when and how patients will be involved into the research process. For instance, how patients are involved in phase 1 to contribute to the design of the care package? Which part of the care package, etc.? How they will be involved into trainings (6-12 p. 10)? Why patients are involved just to do temoignages during trainings while they can have a more active role? For instance, if it is a psycho-social training, they can actually be trainers, for instance describing the challenges patients may face within their care pathway and how the health system/community/family, etc. can support them. They could be trained/act as peer educators, instead of just participating into contact events.*

Response: We agree that it is vitally important to actively include patients (as well as their families and communities) throughout the project. We have now added more details throughout the manuscript in regards to how this will be achieved, in the sections on 'Patient and public involvement' (see pages 13-14), 'Sample selection' (pages 8-9), 'Procedures' (pages 11 and 12), and 'Strengths and limitations' (page 3).

7. *The project foresees also awareness-raising and stigma reduction activities (line 12-15 p. 9). During the design of the care package, some other activities (interviews, focus groups, etc.) can be foreseen to study meanings and experiences around stigma, so to make sure that awareness-raising and stigma reduction activities are based on this and adapted to the needs of those who are stigmatized. Authors should better describe why the project wants to address stigma (is it a barrier to access to health? Or they want to focus on the psycho-social consequences of stigma on affected people, their family, entourage, etc.) and who they want to target (communities? health centres? individuals - patients, caregivers, family members, etc.). This information could help the authors to develop better strategies to reduce stigma and appropriate tools. Moreover, HP stigma-related activities may be conducted by former/current patients instead of health staff: this could be an empowering instrument for patients, instead of reducing them only to activities such as 'temoignages'. Some patient counselling guidelines among the HP materials (line 17 p. 9) are quoted in the protocol. Are they already existing or they should be developed? If yes, will the project contribute to this and how? This could be a very interesting contribution to the NTDs-related research and to national curricula of health workers.*

Response: Thank you for these comments. We agree that the awareness-raising and stigma reduction elements are an important part of the project. We have added further information in various places in the manuscript in regard to the stigma elements of the study, including why it is important to address stigma within the project and who to target (see 'Introduction' section, pages 4-5), and how patients will be more actively involved in the stigma reduction activities (see section on 'Sample selection', pages 8-9). In regards to the patient counselling guidelines, an already existing manual for health workers on psychosocial and economic rehabilitation for patients with podoconiosis and lymphatic filariasis by the National Podoconiosis Action Network (NaPAN) in Ethiopia (2016) was used for this.

8. *Outcomes could be better described and to be more specific. Authors describe how they will measure outcomes/evaluate the care package but they do not explicit them. Is the increased number of patients identified and linked to care an outcome? The decreasing of loss-to-follow up? An improvement of the mental wellbeing of patients? A decrease in reported and experienced stigma in the health centre/in the community? Increased capacity of HWs to reliably diagnose and manage the diseases. This is maybe due to the fact that objectives for each component of the care package may not be very clearly defined.*

Response: Further details about the outcomes and how these are measured have now been included in Table 2 (see pages 24-25).

9. *There are also some other small details which could improve the completeness of the protocol.*

In Figure 3 authors refer to training members of the community. Why we involve them? To improve early case detection? Reduce stigma? Who are them? Opinion leaders? Which are the selection criteria? This should be better clarified.

Response: Thank you for these comments. Figure 3 has now been updated to clarify that these community members would be community leads. Some brief text has also been added in the manuscript, to clarify that these are community leads (see section on 'Sample selection', page 8), that this is a stigma reduction activity (see section on 'Key interventions within the care package', page

10), and that purposive snowballing sampling techniques will be used to select them (see section on 'Sample selection', page 8).

10. *Concerning data management (line 44 p. 10) I would add some more details about where all the electronic fieldnotes, interview/FGDs audio-recordings and transcripts are stored.*

Response: Details in regard to where data are being stored have now been added to the manuscript (see section on 'Ethics', page 15).

11. *Authors could include the potential direct/indirect costs of integrated care for patients and their families into the economic analysis (23-50 p. 11), while not focusing exclusively on the care package costs.*

Response: We have added some details on direct and indirect costs in the manuscript, under 'Economic analysis' (see page 14).

12. *Some more references may be added with reference to integrated care models for NTDs even from other countries, such as Prochazka 2020 in Liberia, Barogui 2020 and Koffi 2020 in Ivory Coast.*

Response: These references have now been added to the manuscript (see page 5).

13. *A timetable may be useful to better understand the different phases of the project.*

Response: A timetable for each of the three phases of the study (in project months) has been added in the 'Study design' section of the manuscript (see page 7).

Reviewer 3

1. *Semrau and co-authors present the protocol for a comprehensive lymphedema prevention and care package integrated into routine health care and targeting three debilitating and stigmatizing neglected tropical diseases. The protocol is well written and the program carefully developed. Below, a couple of comments that might help the authors to further specify certain aspects of the proposed implementation research study.*

Response: We thank the reviewer for these kind comments. We have revised the manuscript based on the reviewer's helpful suggestions below, which we believe has greatly improved the manuscript.

2. *The protocol appears to suggest that lower limb lymphedema were the main consequence of LF, podoconiosis and leprosy. However, particularly in the case of leprosy there are numerous other health problems associated with more severe cases, particularly those with disability grade 2. It is unclear how these will be considered to make the care package truly holistic.*

Response: Thank you for this comment. In Ethiopia there is a well-established leprosy program, which is integrated with the TB program into the national health system. The national leprosy programme targets reducing the proportion of disability grade 2 among newly diagnosed leprosy cases. Nonetheless, morbidity management for lymphoedema secondary to leprosy is minimal. Therefore, the current project addresses a neglected part of leprosy, i.e. lymphoedema secondary to leprosy. Some changes to the wording of the patient target groups have been made throughout the manuscript, i.e. *lower-lymphoedema caused by podocniosis, LF and leprosy*.

3. *Expansion to the whole zone is proposed as the endpoint for the project. To secure knowledge and maximise the impact it seems important to already plan for the time beyond the lifetime of the project (sustainability of intervention) and how findings might be integrated into national policy (policy briefs are mentioned) and practice (roll-out).*

Response: We agree that it would be of great benefit to be able to scale up the care package to the whole of Awi zone, and all endemic districts within Ethiopia. This is not within the remit of the EnDPoINT project, but strategies to ensure sustainability of the intervention include developing the research project in response to a request by the Ministry of Health in Ethiopia, including Team Leads from the NTD Department within the Ministry of Health at all ToC workshops and frequent updates to these Leads through the life of the study.

4. *The project aim is described as “facilitate effective access”. The demonstration of feasibility, acceptability and perception might be emphasized more strongly. It seems appropriate to specifically mention acceptability and perception assessments also in phase 3 (roll-out)*

Response: A table showing the project’s outcomes and associated measures has now been added to the manuscript (see Table 2, pages 24-25 in the marked version of the manuscript), which lists first the implementation outcomes that are being assessed, including feasibility and acceptability. Acceptability as an outcome has now also been highlighted for Phase 3 in Table 1 in the right-hand column (see page 23).

5. *Procedures to accommodate the specific needs of illiterate participants are mentioned for informed consent but not for the study data collection. It is mentioned that English study instruments and materials will be translated into Amharic, but not how they will be made accessible for illiterates. Also, it is not mentioned whether all potential participants speak Amharic as also other languages seem to be common in the region.*

Response: Thank you for these comments. A sentence has now been added in the manuscript, explaining that all measurement tools are being administered by data collectors rather than being completed by participants themselves, so that illiterate participants are not disadvantaged (see ‘Procedures’ section, page 12). Some further text has been included to highlight that Amharic is commonly spoken throughout Ethiopia (see section on ‘Sample selection’, page 9).

6. *The sample size section would benefit from some indications on likely sample sizes that will be targeted/can be expected.*

Response: Details have been added to the ‘Sample size justification’ section of the manuscript in regards to the likely sample sizes that can be expected (see page 9).